# Autoregressive Dynamics Models for Offline Policy Evaluation and Optimization

**Michael R. Zhang**[*1]    **Tom Le Paine**[2]    **Ofir Nachum**[3]    **Cosmin Paduraru**[2]
**George Tucker**[3]    **Ziyu Wang**[3]    **Mohammad Norouzi**[3]

[1]University of Toronto    [2]DeepMind    [3]Google Brain
michael@cs.toronto.edu, mnorouzi@google.com

## Abstract

Standard dynamics models for continuous control make use of feedforward computation to predict the conditional distribution of next state and reward given current state and action using a multivariate Gaussian with a diagonal covariance structure. This modeling choice assumes that different dimensions of the next state and reward are conditionally independent given the current state and action and may be driven by the fact that fully observable physics-based simulation environments entail deterministic transition dynamics. In this paper, we challenge this conditional independence assumption and propose a family of expressive *autoregressive dynamics models* that generate different dimensions of the next state and reward sequentially conditioned on previous dimensions. We demonstrate that autoregressive dynamics models indeed outperform standard feedforward models in log-likelihood on held-out transitions. Furthermore, we compare different model-based and model-free off-policy evaluation (OPE) methods on RL Unplugged, a suite of offline MuJoCo datasets, and find that autoregressive dynamics models consistently outperform all baselines, achieving a new state-of-the-art. Finally, we show that autoregressive dynamics models are useful for offline policy optimization by serving as a way to enrich the replay buffer through data augmentation and improving performance using model-based planning.

## 1 Introduction

Model-based Reinforcement Learning (RL) aims to learn an approximate model of the environment's dynamics from existing logged interactions to facilitate efficient policy evaluation and optimization. Early work on Model-based RL uses simple tabular (Sutton, 1990; Moore and Atkeson, 1993; Peng and Williams, 1993) and locally linear (Atkeson et al., 1997) dynamics models, which often result in a large degree of model bias (Deisenroth and Rasmussen, 2011). Recent work adopts feedforward neural networks to model complex transition dynamics and improve generalization to unseen states and actions, achieving a high level of performance on standard RL benchmarks (Chua et al., 2018; Wang et al., 2019). However, standard feedforward dynamics models assume that different dimensions of the next state and reward are conditionally independent given the current state and action, which may lead to a poor estimation of uncertainty and unclear effects on RL applications.

In this work, we propose a new family of *autoregressive dynamics models* and study their effectiveness for off-policy evaluation (OPE) and offline policy optimization on continuous control. Autoregressive dynamics models generate each dimension of the next state conditioned on previous dimensions of the next state, in addition to the current state and action (see Figure 1). This means that to sample the next state from an autoregressive dynamics model, one needs $n$ sequential steps, where $n$ is the number of state dimensions, and one more step to generate the reward. By contrast, standard feedforward dynamics models take current state and action as input and predict the distribution of the next state and reward as a multivariate Gaussian with a diagonal covariance structure (*e.g.,* Chua et al. (2018); Janner et al. (2019)). This modeling choice assumes that different state dimensions are conditionally independent.

---

*Work done as an intern at Google Brain.

Autoregressive generative models have seen success in generating natural images (Parmar et al., 2018), text (Brown et al., 2020), and speech (Oord et al., 2016), but they have not seen use in Model-based RL for continuous control.

We find that autoregressive dynamics models achieve higher log-likelihood compared to their feed-forward counterparts on heldout validation transitions of all DM continuous control tasks (Tassa et al., 2018) from the RL Unplugged dataset (Gulcehre et al., 2020). To determine the impact of improved transition dynamics models, we primarily focus on OPE because it allows us to isolate contributions of the dynamics model in value estimation *vs.* the many other factors of variation in policy optimization and data collection. We find that autoregressive dynamics models consistently outperform existing Model-based and Model-free OPE baselines on continuous control in both ranking and value estimation metrics. We expect that our advances in model-based OPE will improve offline policy selection for offline RL (Paine et al., 2020). Finally, we show that our autoregressive dynamics models can help improve offline policy optimization by model predictive control, achieving a new state-of-the-art on cheetah-run and fish-swim from RL Unplugged (Gulcehre et al., 2020).

Key contributions of this paper include:

- We propose autoregressive dynamics models to capture dependencies between state dimensions in forward prediction. We show that autoregressive models improve log-likelihood over non-autoregressive models for continuous control tasks from the DM Control Suite (Tassa et al., 2018).
- We apply autoregressive dynamics models to Off-Policy Evaluation (OPE), surpassing the performance of state-of-the art baselines in median absolute error, rank correlation, and normalized top-5 regret across 9 control tasks.
- We show that autoregressive dynamics models are more useful than feedforward models for offline policy optimization, serving as a way to enrich experience replay by data augmentation and improving performance via model-based planning.

## 2 PRELIMINARIES

Here we introduce relevant notation and discuss off-policy (offline) policy evaluation (OPE). We refer the reader to Lange et al. (2012) and Levine et al. (2020) for background on offline RL, which is also known as batch RL in the literature.

A finite-horizon Markov Decision Process (MDP) is defined by a tuple $\mathcal{M} = (\mathcal{S}, \mathcal{A}, \mathcal{T}, d_0, r, \gamma)$, where $\mathcal{S}$ is a set of states $s \in \mathcal{S}$, $\mathcal{A}$ is a set of actions $a \in \mathcal{A}$, $\mathcal{T}$ defines transition probability distributions $p(s_{t+1}|s_t, a_t)$, $d_0$ defines the initial state distribution $d_0 \equiv p(s_0)$, $r$ defines a reward function $r : \mathcal{S} \times \mathcal{A} \to \mathbb{R}$, and $\gamma$ is a scalar discount factor. A policy $\pi(a \mid s)$ defines a conditional distribution over actions conditioned on states. A trajectory consists of a sequence of states and actions $\tau = (s_0, a_0, s_1, a_1, \ldots, s_H)$ of horizon length $H$. We use $s_{t,i}$ to denote the $i$-th dimension of the state at time step $t$ (and similarly for actions). In reinforcement learning, the objective is to maximize the expected sum of discounted rewards over the trajectory distribution induced by the policy:

$$V_\gamma(\pi) = \mathbb{E}_{\tau \sim p_\pi(\tau)} \left[ \sum_{t=0}^H \gamma^t r(s_t, a_t) \right]. \tag{1}$$

The trajectory distribution is characterized by the initial state distribution, policy, and transition probability distribution:

$$p_\pi(\tau) = d_0(s_0) \prod_{t=0}^{H-1} \pi(a_t|s_t) p(s_{t+1}|s_t, a_t). \tag{2}$$

In offline RL, we are given access to a dataset of transitions $\mathcal{D} = \{(s_t^i, a_t^i, r_{t+1}^i, s_{t+1}^i)\}_{i=1}^N$ and a set of initial states $\mathcal{S}_0$. Offline RL is inherently a data-driven approach since the agent needs to optimize the same objective as in Eq. (1) but is not allowed additional interactions with the environment. Even though offline RL offers the promise of leveraging existing logged datasets, current offline RL algorithms (Fujimoto et al., 2019; Agarwal et al., 2020; Kumar et al., 2019) are typically evaluated using online interaction, which limits their applicability in the real world.

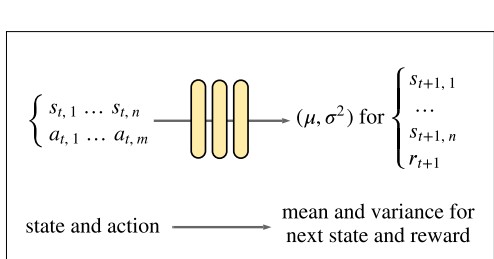
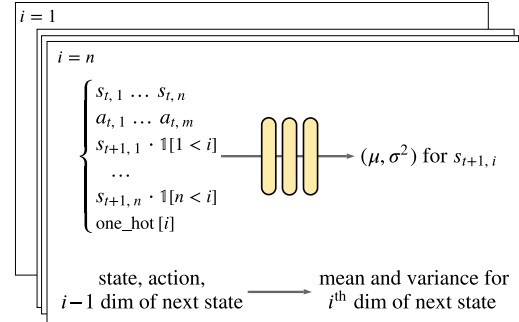

Standard Feedforward Dynamics Models          Proposed Autoregressive Dynamics Model

Figure 1: Standard probabilistic dynamics models (*e.g.,* Chua et al. (2018)) use a neural network to predict the mean and standard deviation of different dimensions of the next state and reward simultaneously. By contrast, we use the same neural network architectures with several additional inputs and predict the mean and standard deviation of each dimension of the next state conditional on previous dimensions of the next state. As empirical results indicate, this small change makes a big difference in the expressive power of dynamics models. Note that reward prediction is not shown on the right to reduce clutter, but it can be thought of as $(n+1)$th state dimension.

The problem of off-policy (offline) policy evaluation (OPE) entails estimating $V_\gamma(\pi)$, the value of a target policy $\pi$, based on a fixed dataset of transitions denoted $\mathcal{D}$, without access to the environment's dynamics. Some OPE methods assume that $\mathcal{D}$ is generated from a known behavior (logging) policy $\mu$ and assume access to $\mu$ in addition to $\mathcal{D}$. In practice, the logged dataset $\mathcal{D}$ may be the result of following some existing system that does not have a probabilistic form. Hence, in our work, we will assume no access to the original behavior policy $\mu$ for OPE. That said, for methods that require access to $\mu$, we train a behavior cloning policy on $\mathcal{D}$.

## 3 PROBABILISTIC DYNAMICS MODELS

**Feedforward dynamics model.** In the context of our paper, we use the term "model" to jointly refer to the forward dynamics model $p_s(s_{t+1}|s_t, a_t)$ and reward model $p_r(r_{t+1}|s_t, a_t)$. We use neural nets to parameterize both distributions since they are powerful function approximators that have been effective for model-based RL (Chua et al., 2018; Nagabandi et al., 2018; Janner et al., 2019).

Let $\theta$ denote the parameters of a fully connected network used to model $p_\theta(s_{t+1}, r_{t+1} \mid s_t, a_t)$. We expect joint modeling of the next state and reward to benefit from sharing intermediate network features. Similar to prior work (Janner et al., 2019), our baseline feedforward model outputs the mean and log variance of all state dimensions and reward simultaneously, as follows:

$$p_\theta(s_{t+1}, r_{t+1} \mid s_t, a_t) \ = \ \mathcal{N}\big(\mu(s_t, a_t), \mathrm{Diag}(\exp\{l(s_t, a_t)\})\big) \,, \tag{3}$$

where $\mu(s_t, a_t) \in \mathbb{R}^{n+1}$ denotes the mean for the concatenation of the next state and reward, $l(s_t, a_t) \in \mathbb{R}^{n+1}$ denotes the log variance, and $\mathrm{Diag}(v)$ is an operator that creates a diagonal matrix with the main diagonal specified by the vector $v$. During training, we seek to minimize the negative log likelihood of the parameters given observed transitions in the dataset $\mathcal{D}$:

$$\ell(\theta \mid \mathcal{D}) \ = \ -\sum\nolimits_{(s,a,r',s')\in\mathcal{D}} \log p_\theta(s', r' \mid s, a) \,. \tag{4}$$

While it is possible to place different weights on the loss for next state and reward prediction, we did not apply any special weighting and treated the reward as an additional state dimension in all of our experiments. This is straightforward to implement and does not require tuning an additional hyperparameter, which is challenging for OPE. Note that the input has $|s| + |a|$ dimensions.

**Autoregressive dynamics model.** We now describe our autoregressive model. We seek to demonstrate the utility of predicting state dimensions in an autoregressive way. Therefore, rather than using a complex neural network architecture, where improvements in log-likelihood and policy evaluation are confounded by architectural differences, we opt to make simple modifications to the feedforward model described above. This allows us to isolate the source of performance improvements.

The autoregressive model we use is a fully connected model that predicts the mean and log variance of a single state dimension. We augment the input space of the baseline with the previous predicted

state dimensions and a one-hot encoding to indicate which dimension to predict. This is illustrated in Figure 1. The autoregressive model therefore has $3|s| + |a|$ input dimensions. Hence, the autoregressive model has a small number of additional weights in the first fully connected layer, but as will be shown in our experiments, these extra parameters are not the reason for a performance gain.

At training time, the autoregressive model has a similar computational cost to the fully connected model as we can mask ground truth states and use data parallelism to compute all state dimensions simultaneously. At inference, the autoregressive model requires additional forward passes, on the order of the number of state dimensions in a given environment. We use the default ordering for the state dimensions in a given environment, though it is interesting to explore different orderings in future works. The negative log-likelihood for an autoregressive model takes the form of:

$$\ell(\theta \mid \mathcal{D}) = -\sum\nolimits_{(s,a,r',s') \in \mathcal{D}} \left[ \log p_\theta(r' \mid s, a, s') + \sum\nolimits_{i=1}^{n} \log p_\theta(s'_i \mid s, a, s'_1, \ldots, s'_{i-1}) \right] , \quad (5)$$

where we use chain rule to factorize the joint probability of $p(s', r' \mid s, a)$.

The main advantage of the autoregressive model is that it makes no conditional independence assumption between next state dimensions. This class of models can therefore capture non-unimodal dependencies, *e.g.,* between different joint angles of a robot. Paduraru (2007) demonstrates this increased expressivity in the tabular setting, constructing an example on which a model assuming conditional independence fails. While the expressive power of autoregressive models have been shown in various generative models (Parmar et al., 2018; Oord et al., 2016), autoregressive dynamics models have not seen much use in Model-based RL for continuous control before this work.

**Model-based OPE.** Once a dynamics model is trained from offline data, OPE can be performed in a direct and primitive way. We let the policy and model interact—the policy generates the next action, the model plays the role of the environment and generates the next state and reward. Due to the stochasticity in the model and the policy, we estimate the return for a policy with Monte-Carlo sampling and monitor standard error. See Algorithm 1 for pseudocode.

---

**Algorithm 1** Model-based OPE

**Require:** Number of rollouts $n$, discount factor $\gamma$, horizon length $H$, policy $\pi$, dynamics model $p$, set of initial states $S_0$
  **for** $i = 1, 2, \ldots n$ **do**
    $R_i \leftarrow 0$
    sample initial state $s_0 \sim \mathcal{S}_0$
    **for** $t = 0, 1, 2, \ldots, H - 1$ **do**
      sample from policy: $a_t \sim \pi(\cdot \mid s_t)$
      sample from the dynamics model:
          $s_{t+1}, r_{t+1} \sim p(\cdot, \cdot \mid s_t, a_t)$
      $R_i \leftarrow R_i + \gamma^t r_{t+1}$
    **end for**
  **end for**
  **return** $\frac{1}{n} \sum_{i=1}^{n} R_i$

---

## 4 RELATED WORK

Our work follows a long line of OPE research, which is especially relevant to many practical domains such as medicine (Murphy et al., 2001), recommendation systems (Li et al., 2011), and education (Mandel et al., 2014) in order to avoid the costs and risks associated with online evaluation. There exists a large body of work on OPE, including methods based on importance weighting (Precup, 2000; Li et al., 2014) and Lagrangian duality (Nachum et al., 2019; Yang et al., 2020; Uehara and Jiang, 2019). The model-based approach that we focus on in this paper lies within the class of algorithms referred to as the *direct method* (Kostrikov and Nachum, 2020; Dudík et al., 2011; Voloshin et al., 2019), which approximate the value of a new policy by either explicitly or implicitly estimating the transition and reward functions of the environment. While model-based policy evaluation has been considered by previous works (Paduraru, 2007; Thomas and Brunskill, 2016a; Hanna et al., 2017), it has largely been confined to simple domains with finite state and action spaces where function approximation is not necessary. By contrast, our work provides an extensive demonstration of model-based OPE in challenging continuous control benchmark domains. Previous instances of the use of function approximation for model-based OPE (Hallak et al., 2015) impose strong assumptions on the probabilistic dynamics models, such as factorability of the MDP. Our results indicate that even seemingly benign assumptions about the independence of different state dimensions can have detrimental consequences for the effectiveness of a model-based OPE estimate.

While the use of model-based principles in OPE has been relatively rare, it has been more commonly used for policy optimization. The field of model-based RL has matured in recent years to yield impressive results for both online (Nagabandi et al., 2018; Chua et al., 2018; Kurutach et al., 2018; Janner et al., 2019) and offline (Matsushima et al., 2020; Kidambi et al., 2020; Yu et al., 2020; Argenson and Dulac-Arnold, 2020) policy optimization. Several of the techniques we employ, such

Table 1: Summary of the offline datasets used. Dataset size indicates the number of $(s, a, r', s')$ tuples.

|  | cartpole swingup | cheetah run | finger turn hard | fish swim | humanoid run | walker stand | walker walk | manipulator insert ball | manipulator insert peg |
|---|---|---|---|---|---|---|---|---|---|
| State dim. | 5 | 17 | 12 | 24 | 67 | 24 | 24 | 44 | 44 |
| Action dim. | 1 | 6 | 2 | 5 | 21 | 6 | 6 | 5 | 5 |
| Dataset size | 40K | 300K | 500K | 200K | 3M | 200K | 200K | 1.5M | 1.5M |

Table 2: Negative log-likelihood on heldout validation sets for different RL Unplugged tasks (lower is better). For both family of dynamics models, we train 48 models with different hyperparameters. We report the Top-1 NLL on the top and average of Top-5 models on the bottom. On all of the tasks autoregressive dynamics models significantly outperform feedforward models in terms of NLL for both Top-1 and Top-5.

|  | Dynamics model architecture | cartpole swingup | cheetah run | finger turn hard | fish swim | humanoid run | walker stand | walker walk | manipulator insert ball | manipulator insert peg |
|---|---|---|---|---|---|---|---|---|---|---|
| Top-1 | Feedforward | -6.81 | -4.90 | -5.58 | -4.91 | -3.42 | -4.52 | -3.84 | -4.74 | -4.34 |
| | Autoregressive | -7.21 | -6.36 | -6.14 | -5.21 | -4.18 | -4.73 | -4.17 | -5.62 | -5.73 |
| Top-5 | Feedforward | -6.75 | -4.85 | -5.50 | -4.90 | -3.40 | -4.49 | -3.81 | -4.64 | -4.31 |
| | Autoregressive | -7.14 | -6.32 | -5.94 | -5.18 | -4.15 | -4.71 | -4.15 | -5.58 | -5.29 |

as the normalization of the observation space, are borrowed from this previous literature (Nagabandi et al., 2018; Chua et al., 2018). Conversely, we present strong empirical evidence that the benefits of our introduced autoregressive generative models of state observations do carry over to model-based policy optimization, at least in the offline setting, and this is an interesting avenue for future work.

## 5 RESULTS

We conduct our experiments on the DeepMind control suite (Tassa et al., 2018), a set of control tasks implemented in MuJoCo (Todorov et al., 2012). We use the offline datasets from RL Unplugged (Gulcehre et al., 2020), the details of which are provided in Table 1. These environments capture a wide range of complexity, from 40K transitions in a 5-dimensional cartpole environment to 1.5 million transitions on complex manipulation tasks. We follow the evaluation protocol in the Deep OPE (Fu et al., 2021) benchmark and use policies generated by four different algorithms: behavioral cloning (Bain, 1995), D4PG (Barth-Maron et al., 2018), Critic Regularized Regression (Wang et al., 2020), and ABM (Siegel et al., 2019). With varied hyperparameters, these form a diverse set of policies of varying quality.

We perform a thorough hyperparameter sweep in the experiments and use standard practice from generative modeling to improve the quality of the models. We allocate 80% of the data for training and 20% of the data for model selection. We vary the depth and width of the neural networks (number of layers $\in \{3, 4\}$, layer size $\in \{512, 1024\}$), add different amounts of noise to input states and actions, and consider two levels of weight decay for regularization (input noise $\in \{0, 1e-6, 1e-7\}$, weight decay $\in \{0, 1e-6\}$). For the choice of optimizer, we consider both Adam (Kingma and Ba, 2014) and SGD with momentum and find Adam to be more effective at maximizing log-likelihood across all tasks in preliminary experiments. We thus use Adam in all of our experiments with two learning rates $\in \{1e-3, 3e-4\}$. We decay the optimizer's learning rate linearly to zero throughout training, finding this choice to outperform a constant learning rate. Lastly, we find that longer training often improves log-likelihood results. We use 500 epochs for training final models.

For each task we consider in total 48 hyperparameter combinations (listed above) for both models and pick the best model in each model family based on validation log-likelihood. This model is then used for model-based OPE and policy optimization. Note that, in our experiments, 20% of the transitions are used only for validation, but we believe one can re-train the models with the best hyperparameter configuration on the full transition datasets to improve the results even further.

### 5.1 AUTOREGRESSIVE DYNAMICS MODELS OUTPERFORM FEEDFORWARD MODELS IN NLL

To evaluate the effectiveness of autoregressive dynamics models compared to feedforward counterparts, Table 2 reports negative log-likelihood (NLL) on the *heldout validation* set for the best

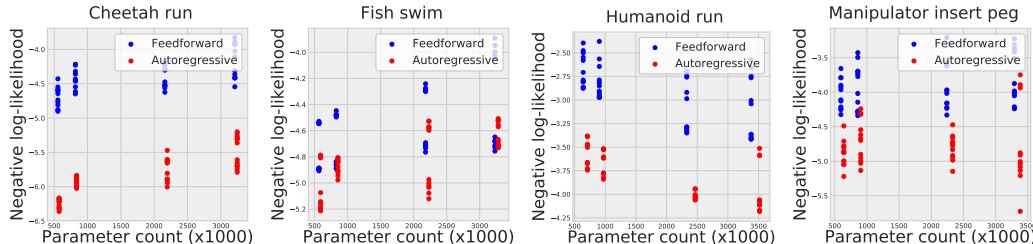

Figure 2: Network parameter count *vs.* validation negative log-likelihood for autoregressive and feedforward models. Autoregressive models often have a lower validation NLL irrespective of parameter count.

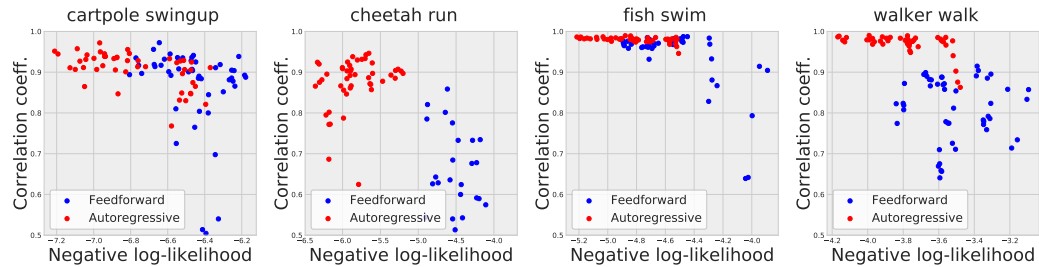

Figure 3: Validation negative log-likelihood *vs.* OPE correlation coefficients on different tasks. On 4 RL Unplugged tasks, we conduct an extensive experiment in which 48 Autoregressive and 48 Feedforward Dynamics models are used for OPE. For each dynamics model, we calculate the correlation coefficient between model-based value estimates and ground truth values at a discount factor of 0.995. We find that low validation NLL numbers generally correspond to accurate policy evaluation, while higher NLL numbers are less meaningful.

performing models from our hyperparameter sweep. For each environment, we report the NLL for the best-performing model (Top-1) and the average NLL across the Top-5 models. The autoregressive model has lower NLL on all environments, indicating that it generalizes better to unseen data.

To study the impact of model size on NLL, Figure 2 shows validation NLL as a function of parameter count. We find that on small datasets large models hurt, but more importantly autoregressive models outperform feedforward models regardless of the parameter count regime, *i.e.,* even small autoregressive models attain a lower validation NLL compared to big feedforward models. This indicates that autoregressive models have a better inductive bias in modeling the transition dynamics than feedforward models that make a conditional independence assumption.

## 5.2   ARE DYNAMICS MODELS WITH LOWER NLL BETTER FOR MODEL-BASED OPE?

We ultimately care not just about the log-likelihood numbers, but also whether or not the dynamics models are useful in policy evaluation and optimization. To study the relationship of NLL and OPE performance for model-based methods, we compute OPE estimates via Algorithm 1 and compute the Pearson correlation between the OPE estimates and the true discounted returns. This serves as a measure of the effectiveness of the model for OPE. We repeat this for all 96 dynamics models we trained on a given environment and plot the correlation coefficients against validation NLL in Figure 3.

Models with low NLL are generally more accurate in OPE. Lambert et al. (2020) have previously demonstrated that in Model-based RL, "training cost does not hold a strong correlation to maximization of episode reward." We use validation NLL instead, and our results on policy evaluation decouple the model from policy optimization, suggesting a more nuanced picture: low validation NLL numbers generally correspond to accurate policy evaluation, while higher NLL numbers are generally less meaningful. In other words, if the dynamics model does not capture the transition dynamics accurately enough, then it is very hard to predict its performance on OPE. However, once the model starts to capture the dynamics faithfully, we conjecture that NLL starts to become a reasonable metric for model selection. For instance, validation NLL does not seem to be a great metric for ranking feedforward models, whereas it is more reasonable for autoregressive models.

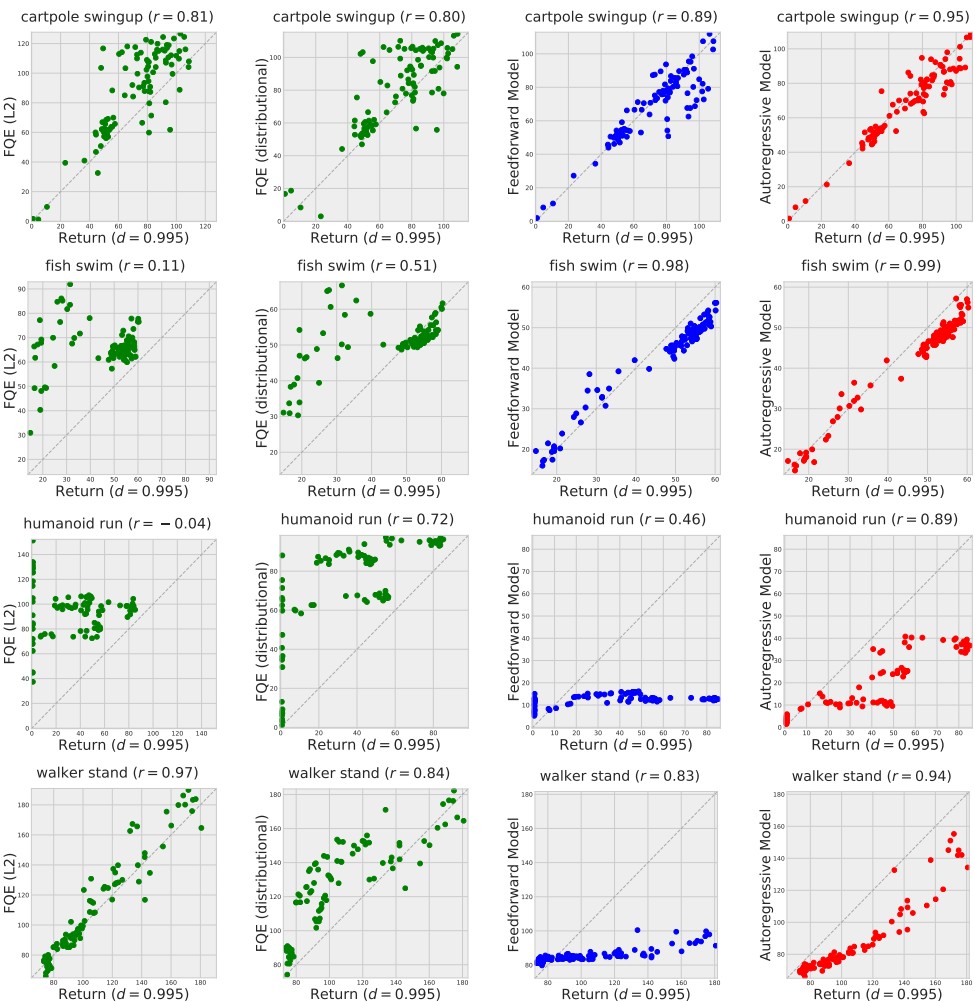

Figure 4: Comparison of model-based OPE using autoregressive and feedforward dynamics models with state-of-the-art FQE methods based on L2 and distributional Bellman error. We plot OPE estimates on the y-axis against ground truth returns with a discount of .995 on the x-axis. We report the Pearson correlation coefficient ($r$) in the title. While feedforward models fall behind FQE on most tasks, autoregressive dynamics models are often superior. See Figure B.4 for additional scatter plots on the other environments.

## 5.3 COMPARISON WITH OTHER OPE METHODS

We adopt a recently proposed benchmark for OPE (Fu et al., 2021) and compare our model-based approaches with state-of-the-art OPE baselines therein. Figures 4 and B.4 compare OPE estimates from two Fitted-Q Evaluation (FQE) baselines (Le et al., 2019; Kostrikov and Nachum, 2020; Paine et al., 2020), our feedforward models, and the autoregressive approach. Each plot reports the Pearson correlation between the OPE estimates and the true returns. The autoregressive model consistently outperforms the feedforward model and FQE methods on most environments. We report ensembling results in the appendix, but compare single models for fairness in the rest of the paper.

We compute summary statistics for OPE methods in Table 3, Table A.1, and Table A.2. These tables report the Spearman's rank correlation, regret, and absolute error, respectively. These metrics capture different desirable properties of OPE methods (Fu et al., 2021); more details about how they are computed are in the appendix. In all three metrics, the autoregressive model achieves the best median performance across nine environments, whereas the baseline model is not as good as FQE. The only environment in which the autoregressive model has negative rank correlation is manipulator insert ball. In addition, a major advantage of our model-based approach over FQE is that the model only needs to be trained once per environment—we do not need to perform additional policy-specific optimization, whereas FQE needs to optimize a separate Q-function approximator per policy.

| | | Cartpole swingup | Cheetah run | Finger turn hard | Fish swim | Humanoid run |
|---|---|---|---|---|---|---|
| Rank Correlation btw. OPE and ground truth | Importance Sampling | $-0.23_{\pm0.11}$ | $-0.01_{\pm0.12}$ | $-0.45_{\pm0.08}$ | $-0.17_{\pm0.11}$ | $\mathbf{0.91}_{\pm\mathbf{0.02}}$ |
| | Best DICE | $-0.16_{\pm0.11}$ | $0.07_{\pm0.11}$ | $-0.22_{\pm0.11}$ | $0.44_{\pm0.09}$ | $-0.10_{\pm0.10}$ |
| | Variational power method | $0.01_{\pm0.11}$ | $0.01_{\pm0.12}$ | $-0.25_{\pm0.11}$ | $0.56_{\pm0.08}$ | $0.36_{\pm0.09}$ |
| | Doubly Robust (IS, FQE) | $0.55_{\pm0.09}$ | $0.56_{\pm0.08}$ | $0.67_{\pm0.05}$ | $0.11_{\pm0.12}$ | $-0.03_{\pm0.12}$ |
| | Feedforward Model | $0.83_{\pm0.05}$ | $\mathbf{0.64}_{\pm\mathbf{0.08}}$ | $0.08_{\pm0.11}$ | $\mathbf{0.95}_{\pm\mathbf{0.02}}$ | $0.35_{\pm0.10}$ |
| | FQE (distributional) | $0.69_{\pm0.07}$ | $\mathbf{0.67}_{\pm\mathbf{0.06}}$ | $\mathbf{0.94}_{\pm\mathbf{0.01}}$ | $0.59_{\pm0.10}$ | $0.74_{\pm0.06}$ |
| | FQE (L2) | $0.70_{\pm0.07}$ | $0.56_{\pm0.08}$ | $0.83_{\pm0.04}$ | $0.10_{\pm0.12}$ | $-0.02_{\pm0.12}$ |
| | Autoregressive Model | $\mathbf{0.91}_{\pm\mathbf{0.02}}$ | $\mathbf{0.74}_{\pm\mathbf{0.07}}$ | $0.57_{\pm0.09}$ | $\mathbf{0.96}_{\pm\mathbf{0.01}}$ | $\mathbf{0.90}_{\pm\mathbf{0.02}}$ |

| | | Walker stand | Walker walk | Manipulator insert ball | Manipulator insert peg | Median ↑ |
|---|---|---|---|---|---|---|
| Rank Correlation btw. OPE and ground truth | Importance Sampling | $0.59_{\pm0.08}$ | $0.38_{\pm0.10}$ | $-0.72_{\pm0.05}$ | $-0.25_{\pm0.08}$ | $-0.17$ |
| | Best DICE | $-0.11_{\pm0.12}$ | $-0.58_{\pm0.08}$ | $0.19_{\pm0.11}$ | $-0.35_{\pm0.10}$ | $-0.11$ |
| | Variational power method | $-0.35_{\pm0.10}$ | $-0.10_{\pm0.11}$ | $\mathbf{0.61}_{\pm\mathbf{0.08}}$ | $\mathbf{0.41}_{\pm\mathbf{0.09}}$ | $0.01$ |
| | Doubly Robust (IS, FQE) | $0.88_{\pm0.03}$ | $0.85_{\pm0.04}$ | $0.42_{\pm0.10}$ | $-0.47_{\pm0.09}$ | $0.55$ |
| | Feedforward Model | $0.82_{\pm0.04}$ | $0.80_{\pm0.05}$ | $0.06_{\pm0.10}$ | $-0.56_{\pm0.08}$ | $0.64$ |
| | FQE (distributional) | $0.87_{\pm0.02}$ | $0.89_{\pm0.03}$ | $\mathbf{0.63}_{\pm\mathbf{0.08}}$ | $-0.23_{\pm0.10}$ | $0.69$ |
| | FQE (L2) | $\mathbf{0.96}_{\pm\mathbf{0.01}}$ | $0.94_{\pm0.02}$ | $\mathbf{0.70}_{\pm\mathbf{0.07}}$ | $-0.48_{\pm0.08}$ | $0.70$ |
| | Autoregressive Model | $\mathbf{0.96}_{\pm\mathbf{0.01}}$ | $\mathbf{0.98}_{\pm\mathbf{0.00}}$ | $-0.33_{\pm0.09}$ | $0.47_{\pm0.09}$ | $0.90$ |

Table 3: Spearman's rank correlation ($\rho$) coefficient (bootstrap mean $\pm$ standard deviation) between different OPE metrics and ground truth values at a discount factor of 0.995. In each column, rank correlation coefficients that are not significantly different from the best ($p > 0.05$) are bold faced. Methods are ordered by median. Also see Table A.1 and Table A.2 for Normalized Regret@5 and Average Absolute Error results.

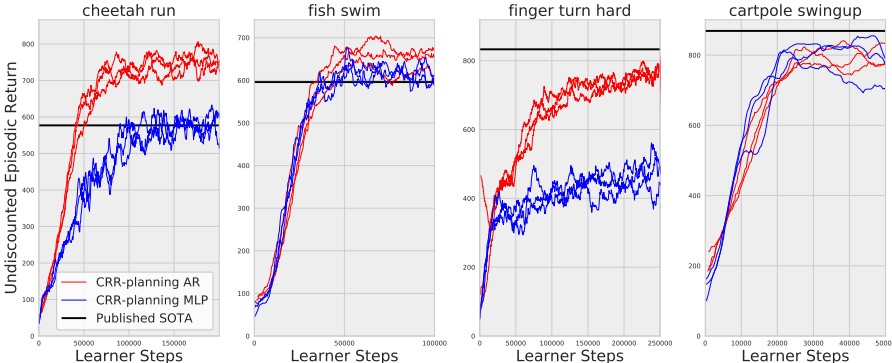

Figure 5: Model-based offline policy optimization results. With planning and data augmentation, we improve the performance over CRR exp (our baseline algorithm). When using autoregressive dynamics models (CRR-planning AR), we outperform state-of-the-art on Cheetah run and Fish swim. Previous SOTA results (Gulcehre et al., 2020; Wang et al., 2020) are obtained using different offline RL algorithms: Cheetah run - *CRR exp*, Fish swim - *CRR binary max*, Finger turn hard - *CRR binary max*, Cartpole swingup - *BRAC* (Wu et al., 2019).

## 5.4 AUTOREGRESSIVE DYNAMICS MODELS FOR OFFLINE POLICY OPTIMIZATION

Policy evaluation is an integral part of reinforcement learning. Improvement in policy evaluation can therefore be adapted for policy optimization. In this section, we explore two possibilities of using models to improve offline reinforcement learning. In all experiments, we use Critic Regularized Regression (CRR) as a base offline reinforcement learning algorithm (Wang et al., 2020).

First, we utilize the model during test time for planning by using a modified version of Model Predictive Path Integral (MPPI) (Williams et al., 2015). Unlike MPPI, we truncate the planning process after 10 steps of rollout and use the CRR critic to evaluate future discounted returns. We provide additional details in the appendix. Secondly, we use the model to augment the transition dataset to learn a better critic for CRR. More precisely, given $s_t^i \sim \mathcal{D}$, and the current policy $\pi$, we can generate additional data using the following process: $\hat{a}_t^i \sim \pi(\cdot|s_t^i), \quad \hat{s}_{t+1}^i, \hat{r}_{t+1}^i \sim p(\cdot, \cdot|s_t^i, \hat{a}_t)$.

These two options are orthogonal and can be applied jointly. We implemented both techniques on top of the CRR *exp* variant (Wang et al., 2020) and show their combined effect in Figure 5. The

figure shows that autoregressive dynamics models also outperform feedforward ones in the policy optimization context. Notably, in the case of cheetah run and fish swim, using autoregressive models for planning as well as data augmentation enables us to outperform the previous state-of-the-art on these offline datasets. Additionally, when using autoregressive dynamics models, both techniques improve performance. In the appendix, we show this result as well as more ablations.

## 6 CONCLUSION

This paper shows the promise of autoregressive models in learning transition dynamics for continuous control, showing strong results for off-policy policy evaluation and offline policy optimization. Our contributions to offline model-based policy optimization are orthogonal to prior work that uses ensembles to lower the values when ensemble components disagree (Kidambi et al., 2020). Incorporating conservative value estimation into our method is an interesting avenue for future research. We use relatively primitive autoregressive neural architectures in this paper to enable a fair comparison with existing feedforward dynamics models. That said, it will be exciting to apply more sophisticated autoregressive neural network architectures with cross attention (Bahdanau et al., 2014) and self-attention (Vaswani et al., 2017) to Model-based RL for continuous control.

**Acknowledgements** We thank Jimmy Ba, William Chan, Rishabh Agarwal, Dale Schuurmans, and Silviu Pitis for fruitful discussions on our work. We are also grateful for the helpful comments from Lihong Li, Jenny Liu, Harris Chan, Keiran Paster, Sheng Jia, and Tingwu Wang on earlier drafts.

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

# A    OFFLINE POLICY EVALUATION

We use the baseline results in Fu et al. (2021). For convenience, we replicate their description of the OPE baselines and metrics.

## A.1    OPE METRICS

To evaluate the OPE algorithms, we compute three different metrics between the estimated returns and the ground truth returns:

1. **Rank correlation** This metric assesses how well estimated values rank policies. It is equal to the correlation between the ranking (sorted order) by the OPE estimates and the ranking by the ground truth values.

2. **Absolute Error**: This metric measures the deviations of the estimates from the ground truth and does not directly access the usefulness for ranking.

3. **Regret@k** This metric measures how much worse the best policies identified by the estimates are than the best policy in the entire set. Regret@k is the difference between the actual expected return of the best policy in the entire set, and the actual value of the best policy in the top-k set.

## A.2    OPE BASELINES

**Fitted Q-Evaluation (FQE)** As in Le et al. (2019), we train a neural network to estimate the value of the evaluation policy $\pi_e$ by bootstrapping from $Q(s', \pi_e(s'))$. We tried two different implementations, one from Kostrikov and Nachum (2020) and another from Paine et al. (2020).

**Importance Sampling (IS)** We perform importance sampling with a learned behavior policy. We use the implementation from Kostrikov and Nachum (2020), which uses self-normalized (also known as weighted) step-wise importance sampling (Liu et al., 2018; Nachum et al., 2019). Since the behavior policy is not known explicitly, we learn an estimate of it via a max-likelihood objective over the dataset $\mathcal{D}$, as advocated by Hanna et al. (2019). In order to be able to compute log-probabilities when the target policy is deterministic, we add artificial Gaussian noise with standard deviation 0.01 for all deterministic target policies.

**Doubly-Robust (DR)** We perform weighted doubly-robust policy evaluation based on Thomas and Brunskill (2016b) and using the implementation of Kostrikov and Nachum (2020). Specifically, this method combines the IS technique above with a value estimator for variance reduction. The value estimator is learned according to Kostrikov and Nachum (2020), using deep FQE with an L2 loss function.

**DICE** This method uses a saddle-point objective to estimate marginalized importance weights $d^\pi(s, a)/d^{\pi_B}(s, a)$; these weights are then used to compute a weighted average of reward over the offline dataset, and this serves as an estimate of the policy's value in the MDP. We use the implementation from Yang et al. (2020) corresponding to the algorithm *BestDICE*.

**Variational Power Method (VPM)** This method runs a variational power iteration algorithm to estimate the importance weights $d^\pi(s, a)/d^{\pi_B}(s, a)$ without the knowledge of the behavior policy. It then estimates the target policy value using weighted average of rewards similar to the DICE method. Our implementation is based on the same network and hyperparameters for OPE setting as in Wen et al. (2020). We further tune the hyperparameters including the regularization parameter $\lambda$, learning rates $\alpha_\theta$ and $\alpha_v$, and number of iterations on the Cartpole swingup task using ground-truth policy value, and then fix them for all other tasks.

## A.3    ENSEMBLING

As in Chua et al. (2018); Janner et al. (2019), we can form an ensemble using our best-performing models. We generate rollouts using the procedure detailed in Janner et al. (2019), forming an ensemble with 4 models. We see some improvement in policy evaluation results, as shown in Figure A.1. Ensembling could likely be further improved by forcing unique hyperparameter settings and seeds.

| | | Cartpole swingup | Cheetah run | Finger turn hard | Fish swim | Humanoid run |
|---|---|---|---|---|---|---|
| **Regret@5** for OPE *vs.* ground truth | Importance Sampling | 0.73 ±0.16 | 0.40 ±0.21 | 0.64 ±0.05 | 0.12 ±0.05 | **0.31** ±0.09 |
| | Best DICE | 0.68 ±0.41 | 0.27 ±0.05 | 0.44 ±0.04 | **0.35** ±0.24 | 0.84 ±0.22 |
| | Variational power method | 0.50 ±0.13 | 0.37 ±0.04 | 0.45 ±0.13 | **0.02** ±0.02 | 0.56 ±0.08 |
| | Doubly Robust (IS, FQE) | 0.28 ±0.05 | **0.09** ±0.05 | 0.56 ±0.12 | 0.61 ±0.12 | 0.99 ±0.00 |
| | FQE (L2) | 0.06 ±0.04 | 0.17 ±0.05 | **0.30** ±0.11 | 0.50 ±0.03 | 0.99 ±0.00 |
| | Feedforward Model | **0.02** ±0.02 | 0.24 ±0.12 | 0.43 ±0.04 | **0.00** ±0.00 | 0.44 ±0.02 |
| | FQE (distributional) | 0.03 ±0.09 | 0.11 ±0.09 | 0.10 ±0.12 | 0.49 ±0.06 | **0.24** ±0.15 |
| | Autoregressive Model | **0.00** ±0.02 | **0.01** ±0.02 | 0.63 ±0.11 | **0.03** ±0.02 | **0.32** ±0.06 |
| | | **Walker stand** | **Walker walk** | **Manipulator insert ball** | **Manipulator insert peg** | **Median ↓** |
| **Regret@5** for OPE *vs.* ground truth | Importance Sampling | 0.54 ±0.11 | 0.54 ±0.23 | 0.83 ±0.05 | **0.22** ±0.03 | 0.54 |
| | Best DICE | 0.24 ±0.07 | 0.55 ±0.06 | **0.44** ±0.07 | 0.75 ±0.04 | 0.44 |
| | Variational power method | 0.41 ±0.02 | 0.39 ±0.02 | **0.52** ±0.20 | 0.32 ±0.02 | 0.41 |
| | Doubly Robust (IS, FQE) | **0.02** ±0.01 | **0.05** ±0.07 | **0.30** ±0.10 | 0.73 ±0.01 | 0.30 |
| | FQE (L2) | **0.04** ±0.02 | **0.00** ±0.02 | **0.37** ±0.07 | 0.74 ±0.01 | 0.30 |
| | Feedforward Model | 0.18 ±0.10 | **0.03** ±0.05 | 0.83 ±0.06 | 0.74 ±0.01 | 0.24 |
| | FQE (distributional) | **0.03** ±0.03 | **0.01** ±0.02 | 0.50 ±0.30 | 0.73 ±0.01 | 0.11 |
| | Autoregressive Model | **0.04** ±0.02 | **0.04** ±0.02 | 0.85 ±0.02 | **0.30** ±0.04 | 0.04 |

Table A.1: Normalized Regret@5 (bootstrap mean ± standard deviation) for OPE methods *vs.* ground truth values at a discount factor of 0.995. In each column, normalized regret values that are not significantly different from the best ($p > 0.05$) are bold faced. Methods are ordered by median.

| | | Cartpole swingup | Cheetah run | Finger turn hard | Fish swim | Humanoid run |
|---|---|---|---|---|---|---|
| **Absolute Error** btw. OPE and ground truth | Variational power method | 37.53 ±3.50 | 61.89 ±4.25 | 46.22 ±3.93 | 31.27 ±0.99 | 35.29 ±3.03 |
| | Importance Sampling | 68.75 ±2.39 | 44.29 ±1.91 | 90.10 ±4.68 | 34.82 ±1.93 | 27.89 ±1.98 |
| | Best DICE | 22.73 ±1.65 | 23.35 ±1.32 | 33.52 ±3.48 | 59.48 ±2.47 | 31.42 ±2.04 |
| | Feedforward Model | **6.80** ±0.85 | 13.64 ±0.59 | 35.99 ±3.00 | **4.75** ±0.23 | 30.12 ±2.40 |
| | FQE (L2) | 19.02 ±1.34 | 48.26 ±1.78 | 27.91 ±1.18 | 19.82 ±1.57 | 56.28 ±3.52 |
| | Doubly Robust (IS, FQE) | 24.38 ±2.51 | 40.27 ±2.05 | 25.26 ±2.48 | 20.28 ±1.90 | 53.64 ±3.68 |
| | FQE (distributional) | 12.63 ±1.21 | 36.50 ±1.62 | **10.23** ±0.93 | 7.76 ±0.95 | 32.36 ±2.27 |
| | Autoregressive Model | **5.32** ±0.54 | **4.64** ±0.46 | 22.93 ±1.72 | **4.31** ±0.22 | **20.95** ±1.61 |
| | | **Walker stand** | **Walker walk** | **Manipulator insert ball** | **Manipulator insert peg** | **Median ↓** |
| **Absolute Error** btw. OPE and ground truth | Variational power method | 96.76 ±3.59 | 87.24 ±4.25 | 79.25 ±6.19 | 21.95 ±1.17 | 46.22 |
| | Importance Sampling | 66.50 ±1.90 | 67.24 ±2.70 | 29.93 ±1.10 | 12.78 ±0.66 | 44.29 |
| | Best DICE | 27.58 ±3.01 | 47.28 ±3.13 | 103.45 ±5.21 | 22.75 ±3.00 | 31.42 |
| | Feedforward Model | 23.34 ±2.41 | 52.23 ±2.34 | 34.30 ±2.55 | 121.12 ±1.58 | 30.12 |
| | FQE (L2) | **6.51** ±0.71 | 18.34 ±0.95 | 36.32 ±1.07 | 31.12 ±2.37 | 27.91 |
| | Doubly Robust (IS, FQE) | 26.82 ±2.66 | 24.63 ±1.69 | 13.33 ±1.16 | 22.28 ±2.34 | 24.63 |
| | FQE (distributional) | 21.49 ±1.41 | 27.57 ±1.54 | **9.75** ±1.10 | 12.66 ±1.39 | 12.66 |
| | Autoregressive Model | 19.12 ±1.23 | **5.14** ±0.49 | 17.13 ±1.34 | **9.71** ±0.70 | 9.71 |

Table A.2: Average absolute error between OPE metrics and ground truth values at a discount factor of 0.995. In each column, absolute error values that are not significantly different from the best ($p > 0.05$) are bold faced. Methods are ordered by median.

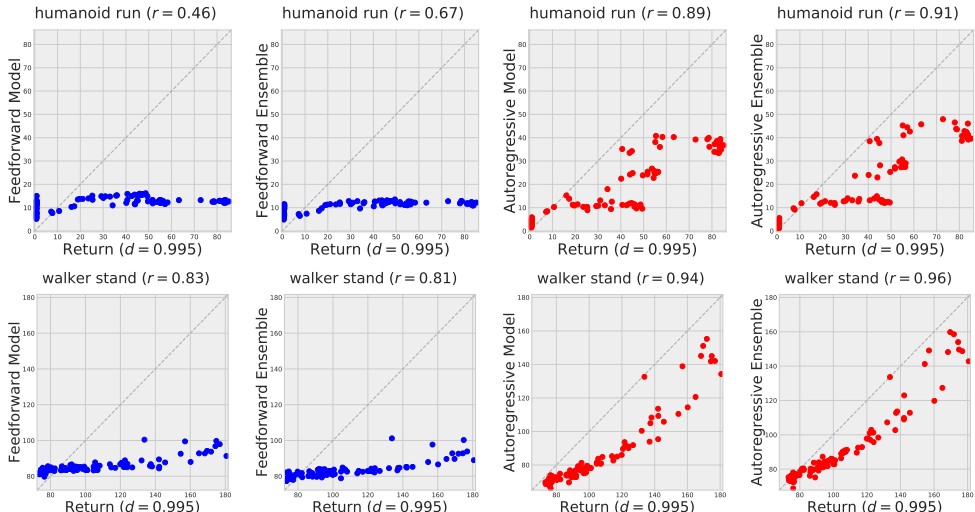

Figure A.1: Estimates of returns using the top model versus estimates of returns using an ensemble of the top-4 models.

---

**Algorithm 2** Model Predictive Path Integral Planning

---

**Require:** state $s$, policy $\pi$, dynamics model $p$, critic $Q$, temperature $\beta$, and noise variance $\sigma^2$.
  **for** $m = 1, ..., M$ **do**
    **for** $n = 1, ..., N$ **do**
      $s_0 \leftarrow s$
      $R_n \leftarrow 0$
      **for** $\tau = 0, ..., H-1$ **do**
        $a_n^\tau \sim \pi(\cdot | s_n^\tau)$
        $s_n^{\tau+1}, r_n^{\tau+1} \sim \pi(\cdot, \cdot | s_n^\tau, a_n^\tau)$
        $R_n \leftarrow R_n + \gamma^\tau r_n^{\tau+1}$
      **end for**
      $a^H \sim \pi(\cdot | s^H)$
      $R_n \leftarrow R_n + \gamma^H Q(s_n^H, a_n^H)$
    **end for**
    Re-define $\pi$ such that $\pi(\cdot | \hat{s}^\tau) = \sum_n \frac{\exp(R_n/\beta)}{\sum_m \exp(R_m/\beta)} \mathcal{N}(\cdot | a_n^\tau, \sigma^2 I)$. ($\pi$ depends on $\tau$ and not $\hat{s}$.)
  **end for**
  sample final action $a \sim \sum_n \frac{\exp(R_n/\beta)}{\sum_m \exp(R_m/\beta)} \delta(a_n^0)$
  **return** $a$

---

# B ADDITIONAL DETAILS REGARDING POLICY OPTIMIZATION

To test dynamic models for policy optimization, we implement the two methods discussed in Section 5.4 on top of CRR *exp*, one of the CRR variants (Wang et al., 2020). We use the RL Unplugged datasets (Gulcehre et al., 2020) for all environments studied in this section. When using data augmentation, we adopt a 1-to-1 ratio between the original dataset and the augmented dataset.

To take advantage of the dynamics models at test time, we use a variant of Model Predictive Path Integral (MPPI) for planning. To reduce the planning horizon, we truncate the model rollout using CRR critics. The details of the planning procedure is summarized in Algorithm 2. All hyperparameter tuning for the planning process is conducted on the "cartpole swingup" task. The hyperparameters used in the planning process are $M = 3, N = 16, H = 10, \beta = 0.1$, and $\sigma^2 = 0.01$. To match the temperature used in the planning component, we choose $\beta = 0.1$ for the CWP component of CRR. This change, however, does not impact the baseline CRR agent performance much. With the exception of $\beta$ and the planning component, all hyperparameters are kept the same as CRR *exp*.

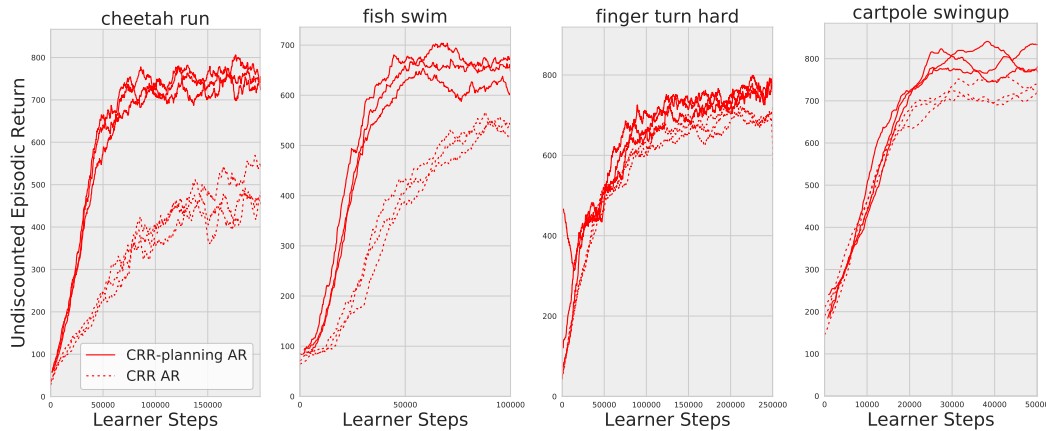

Figure B.2: Effects of the planning procedure. Here we compare using planning (CRR-planning AR) vs not ((CRR AR)) while using augmented data generated by the autoregressive model. Planning with autoregressive models helps in all environments tested.

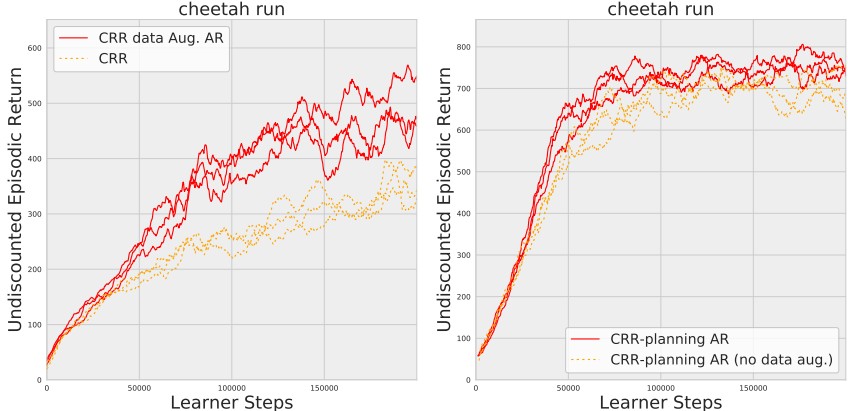

Figure B.3: Effects of the data augmentation on cheetah run. [LEFT] In the absence of planning, data augmentation significantly increase the performance of CRR agent. [RIGHT] With the planning procedure, data augmentation is still effective albeit to a lesser extent.

We compare the agents' performance with and without the planning procedure to test its effects. As shown in Figure B.2, planning using an autoregressive model significantly increases performance.

Data augmentation does not change the agents' performance on cartpole swingup, fish swim, or finger turn hard. It, however, boosts performance considerably on cheetah run. In Figure B.3, we show the effects of data augmentation on cheetah run.

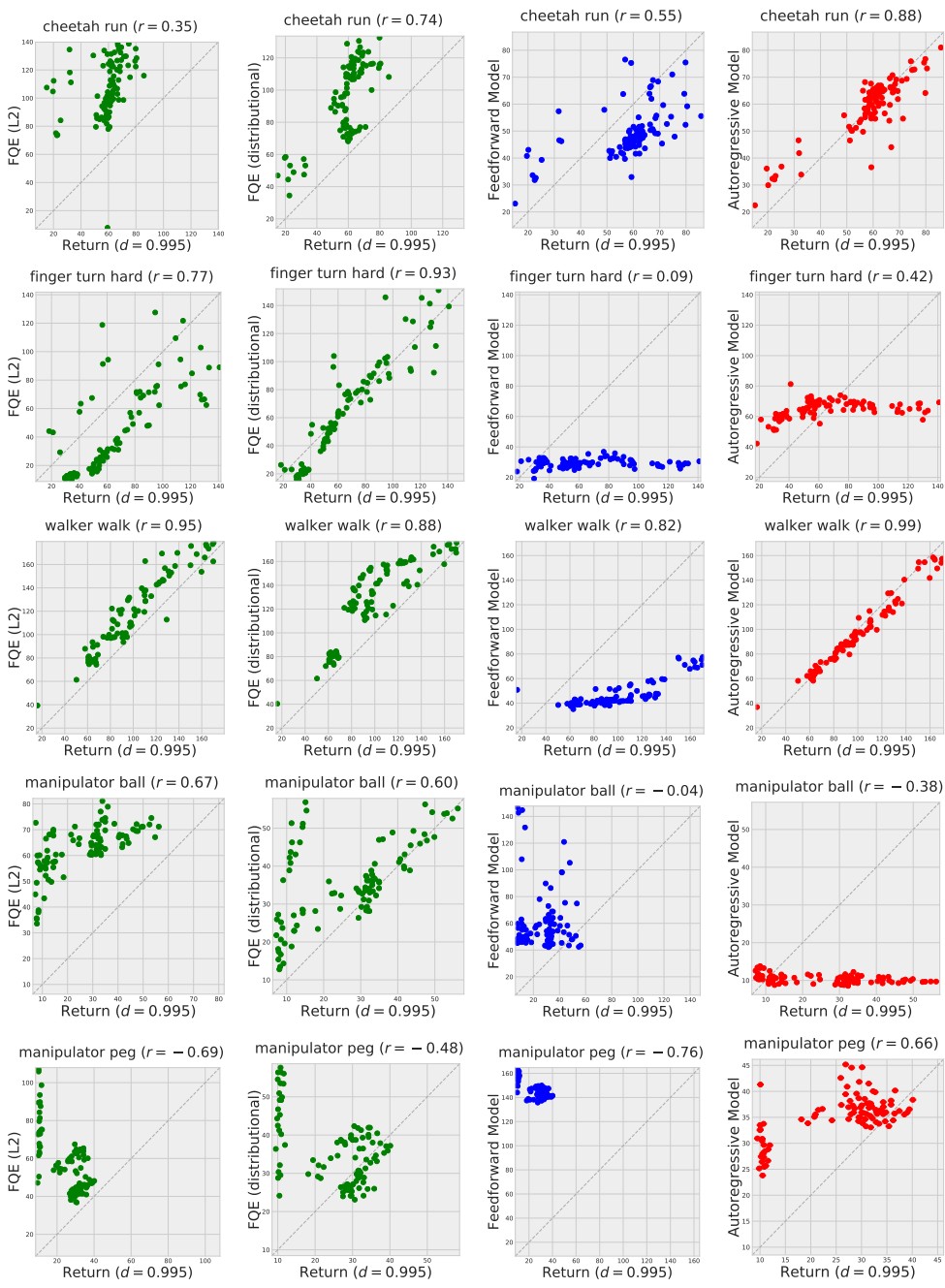

Figure B.4: Comparison of model-based OPE using autoregressive and feedforward dynamics models with state-of-the-art FQE methods based on L2 and distributional Bellman error. We plot ground truth returns on the x-axis against estimates of returns from various OPE methods on the y-axis. While feedforward models fall behind FQE on most tasks, autoregressive dynamics models are often superior. The remaining environments are plotted in Figure 4.

