# OpenReview forum: "Autoregressive Dynamics Models for Offline Policy Evaluation and Optimization"
_ICLR.cc/2021/Conference — ICLR 2021 Poster_

### Official Review · AnonReviewer3 · 2020-10-23
**Good results by applying autoregressive dynamics models to batch policy evaluation/optimization**

**Rating:** 7
**Confidence:** 3

**Review:**

#### Summary

The authors consider the usage of autoregressive dynamics models for batch model-based RL, where state-variable/reward predictions are performed sequentially conditioned on previously-predicted variables. Extensive numerical results are provided in several continuous domains for both policy evaluation and optimization problems. The results showcase the effectiveness of autoregressive models and, in particular, their superiority over standard feed-forward models.

#### Pros

- The paper is very well-written and easy to follow. The experiments are described with sufficient details to understand the results
- The usage of these autoregressive models for model-based RL is, to my knowledge, novel
- The paper presents extensive experiments on several challenging domains. The results are convincing and significant. In particular, they show that autoregressive models are superior to feedforward ones

#### Cons

- The paper's sole contribution seems to be empirical since autoregressive models are (as acknowledged) not novel, though their application to this setting is.
- While the empirical results are very convincing, I did not find much intuition on where this big improvement over feedforward models comes from (see detailed comments below).
- The ordering of the state variables might be a limitation (again, see below).

#### Detailed comments

1. As mentioned above, I did not find much intuition on the better performances of autoregressive models vs feedforward ones. As I am not entirely familiar with the system dynamics of the considered domains, do you think that they possess any property which makes autoregressive models more suitable than feedforward ones (e.g., strong correlations between next-state variables)? Aren't the transition dynamics deterministic in most of the considered domains?

2. Since the reward in most of the considered domains is (I suppose) a function of state, action, and next-state, could it be that one of the reasons behind the worse performance of feedforward models is that they try to predict the reward as a function of state-action only? Would their performance change if they explicitly modeled the reward as a function of s,a,s'?

3. Related to the previous point, the autoregressive model naturally predicts a reward as a function of s,a,s' since r is considered as the (n+1)-th state component. But what if we re-ordered the state variables with r as the first component instead of the last one? Would the performance change?

4. More generally, do you think that the ordering of the state variables might be a limitation? For instance, could there be an ordering of these variables that makes the model perform well and one that makes it perform poorly? While in, e.g., image/text generation problems where autoregressive models are applied we have a natural ordering between the variables involved (e.g., by space or time), here there seems to be no particular relationship between state variables with similar index. Maybe some additional experiments could help in clarifying whether this could be a limitation or not.

Some minor comments/questions:

- In Eq. 2, should the product be up to H-1?
- Before Sec. 3, a citation for "behavioral cloning" could be added
- Sec. 5.3: the FQE acronym was not introduced
- Fig. 4: what is "r" above each plot?

---

> ### Author Response · Authors · 2020-11-20
> **Response to reviewer 3**
>
> Thank you for your valuable feedback and suggestions!
>
> >As mentioned above, I did not find much intuition on the better performances of autoregressive models vs feedforward ones. As I am not entirely familiar with the system dynamics of the considered domains, do you think that they possess any property which makes autoregressive models more suitable than feedforward ones (e.g., strong correlations between next-state variables)? Aren't the transition dynamics deterministic in most of the considered domains?
>
> The environments are indeed deterministic. Nevertheless, explicitly modeling the conditional dependence between state dimensions achieves better performance. This can stem from the fact that in DM Control, each task constitutes certain angle constraints for corresponding characters. Standard feedforward models with diagonal covariance structure have no way of ensuring that these angle constraints are satisfied when there is a lot of uncertainty in the prediction. On the other hand, the autoregressive dynamics model is able to avoid unacceptable configurations of the character and ensure angles are compatible with each other. We will add an analysis and an explanation to this phenomenon to the paper.
>
> >Since the reward in most of the considered domains is (I suppose) a function of state, action, and next-state, could it be that one of the reasons behind the worse performance of feedforward models is that they try to predict the reward as a function of state-action only? Would their performance change if they explicitly modeled the reward as a function of s,a,s'? [...] r is considered as the (n+1)-th state component. But what if we re-ordered the state variables with r as the first component instead of the last one? Would the performance change?
>
> As explained above, we believe the autoregressive model has a much higher expressive power than the two-stage model that you proposed. That said, we agree that the two-stage dynamics model is a useful baseline, which time-permitting we will include in the final paper. We are in the process of finalizing experiments with autoregressive models using a reverse order of [state dimensions, reward], which to some extent addresses your first concern too.
>
> >More generally, do you think that the ordering of the state variables might be a limitation? For instance, could there be an ordering of these variables that makes the model perform well and one that makes it perform poorly? While in, e.g., image/text generation problems where autoregressive models are applied we have a natural ordering between the variables involved (e.g., by space or time), here there seems to be no particular relationship between state variables with similar index. Maybe some additional experiments could help in clarifying whether this could be a limitation or not.
>
> We believe the order of  [state dimensions, reward] does not have a significant impact on performance given prior work in NLP, which studies the effect of generation order and finds little difference between different generation orders. For instance, Chan et al. (2019) show for machine translation, unintuitive orderings such as alphabetical can match the performance of the standard left-to-right order. It appears that modeling conditional dependencies explicitly is the most important component, since any order can be used to re-express the joint probability using chain rule.
>
>
> >In Eq. 2, should the product be up to H-1?
>
> Yes, thank you.
>
> > Before Sec. 3, a citation for "behavioral cloning" could be added
>
> Done, thanks.
>
> > Sec. 5.3: the FQE acronym was not introduced
>
> Fixed, thanks.
>
> > Fig. 4: what is "r" above each plot?
>
> r refers to r-value, which is the Pearson correlation coefficient. We made this more explicit in the caption in the revision.
>
>
> Chan, William, et al. "An Empirical Study of Generation Order for Machine Translation." arXiv preprint arXiv:1910.13437 (2019).

---

> > ### Comment · AnonReviewer3 · 2020-11-24
> > **Response to authors' feedback**
> >
> > Dear authors,
> >
> > Thank you for your detailed response. That addresses my comments. I have increased my score accordingly.

---

### Official Review · AnonReviewer4 · 2020-10-27
**Autoregressive structure that predicts each coordinate of next state sequentially instead of all coordinates simultaneously helps for better offline RL tasks such as OPE**

**Rating:** 6
**Confidence:** 4

**Review:**

The paper proposes extra conditioning in a dynamics model, wherein each dimension of the next state is generated on previous dimensions as well as previous state and action. This allows for a richer model (as a model without conditioning on previous dimensions is a special case).The paper claims that this additional conditioning adds a better inductive bias in certain tasks.


The new models fit data better (Deepmind suite, RL Unplugged data) in terms of log-likelihood. The authors study the impact of better models on OPE and on policy optimization. The paper is generally well written. The contribution seems straightforward.


Why is this relevant only in continuous control tasks? Is this inductive bias really a general pattern observed in multiple tasks? What happens when there is no structure a priori (e.g., independence holds): do you lose in terms of sample efficiency?

Shouldn't any dynamics model be able to enrich replay buffer?

The default ordering may be completely arbitrary, how is the new dynamics model able to cope with this in the experiments? In other words, it is unclear if P(s_i|s_{j<i},...) is appropriate at all without knowing the ordering.

Also since p(s|s_prev,a) = \Prod_{i}p(s_i|s_prev,a,s_{j<i}) by definition, how is the feedforward model restrictive and making explicit conditional independence assumption? It seems that the feedforward model is not restrictive but too general and explicitly capturing this sequential dependency across states is useful.

>> We ultimately care not about the log-likelihood numbers, but whether or not the dynamics models are faithful in policy evaluation and optimization.

The above needs clarification. What is the faithfulness property? Also, why would we not care about log-likelihood?

---

> ### Author Response · Authors · 2020-11-20
> **Response to reviewer 2**
>
> Thank you for your valuable feedback! We address your concerns:
>
> >Why is this relevant only in continuous control tasks? Is this inductive bias really a general pattern observed in multiple tasks?
>
> The DM control suite tasks that we consider comprise a variety of tasks with different levels of difficulties, from cartpole with 5-dimensional states to humanoid with 67-dimensional states. So, we do believe that we have shown enough evidence that the inductive bias of autoregressive models is beneficial to a large family of tasks. Given previous work on autoregressive generative models of text (e.g., GPT-3), images (e.g., PixelCNN), and audio (e.g., WaveNet) showing state-of-the-art results, we believe autoregressive dynamics models would also perform well in other domains, e.g., discrete control. We constrained the scope of the paper to continuous control because continuous control benchmarks are popular and feedforward models with a diagonal covariance matrix represent the state-of-the-art in these domains.
>
> >What happens when there is no structure a priori (e.g., independence holds): do you lose in terms of sample efficiency?
>
> In general, independence *does* hold in the environments that we used, given that the physics-based simulator is deterministic. Since we tackle policy optimization and evaluation in the offline setting, all of the approaches evaluated use the same number of samples -- i.e., they are equally sample-efficient.
>
> >Shouldn't any dynamics model be able to enrich replay buffer?
>
> Yes, any dynamics model can be used to generate synthetic transitions to augment the replay buffer. That said, if the quality of synthetic transitions is poor, it can hurt policy optimization. In our experiments we find that the use of synthetic transitions generated by a feedforward dynamics model provides a much smaller improvement compared to an autoregressive dynamics model (See Figure 5 and B.3).
>
>
> >The default ordering may be completely arbitrary, how is the new dynamics model able to cope with this in the experiments? In other words, it is unclear if P(s_i|s_{j<i},...) is appropriate at all without knowing the ordering.
>
> Agreed. However, any ordering can help model the conditional dependencies between state dimensions that are completely ignored by a standard feedforward model with a diagonal covariance structure. We will include experimental results with the reverse order of state dimensions in the final version of the paper.
>
> >Also since p(s|s_prev,a) = \Prod_{i}p(s_i|s_prev,a,s_{j<i}) by definition, how is the feedforward model restrictive and making explicit conditional independence assumption? It seems that the feedforward model is not restrictive but too general and explicitly capturing this sequential dependency across states is useful.
>
> We are referring to feedforward dynamics models for continuous control that parametrize the next state as a Gaussian with a diagonal covariance structure (Chua et al. (2018); Janner et al. (2019)). In such a parameterization, each dimension of the next state is conditionally independent given the current state and action. We have made this more clear in an updated introduction and agree with your point that explicitly capturing this sequential dependency is useful.
>
> >We ultimately care not about the log-likelihood numbers, but whether or not the dynamics models are faithful in policy evaluation and optimization.
> >>The above needs clarification. What is the faithfulness property? Also, why would we not care about log-likelihood?
>
> Thank you for this suggestion; we clarify the sentence in our revision. Our intention was to say that we care not just about log-likelihood, but about how useful the models are in improving performance on downstream tasks e.g., policy evaluation and improvement. Log-likelihood numbers alone are not useful if they do not correspond meaningfully to downstream tasks. Lambert et al. observed in the online setting that training log-likelihood alone does not correspond to meaningful improvements. We use validation NLL in the offline setting, and we show that low NLL generally corresponds to accurate policy evaluation (high correlation between ground-truth and model-based estimates of returns).
>
> Lambert, Nathan, et al. "Objective Mismatch in Model-based Reinforcement Learning." arXiv preprint arXiv:2002.04523 (2020).
>
> Please let us know if additional clarifications are needed.

---

### Official Review · AnonReviewer1 · 2020-10-29
**Simple extension, convincing empirical results**

**Rating:** 7
**Confidence:** 4

**Review:**

Summary

The paper studies offline policy evaluation (OPE) and optimization in the model-based setting. The main methodological contribution of the paper is using autoregressive models for the next state and reward prediction. The authors demonstrate that autoregressive models achieve higher likelihood compared to feedforward models on 9 environments from RL Unplugged [1] offline dataset. Given that model likelihood is only a proxy quality metric in OPE and control, they further demonstrate a positive correlation between likelihood and OPE estimates. The paper shows quantitatively that using autoregressive models results in more accurate OPE estimates than for feedforward models and model-free benchmarks. Finally, the authors apply autoregressive models for offline control and achieve higher returns than for feedforward models.


Strengths
- The paper is written clearly and generally easy to follow.
- The proposed modification is simple, straightforward to implement, and demonstrates convincing results consistently on different environments. For example, the median rank correlation between OPE and ground truth is the best against 7 OPE baselines on 9 environments from RL Unplugged.
- The experimental setup follows the standard practices (e.g. using a validation set for hyperparameter selection) and the details necessary for the reproduction of the results are provided (e.g. optimizers, learning rate schedules, number of epochs, architectures).

Weaknesses
- The authors claim that “standard feedforward dynamics models assume that different dimensions of the next state and reward are conditionally independent given the current state and action”. In other words, p(s’,r|s,a) is claimed to be equal to p(s’_1|s,a) … p(s’_n|s,a) p(r|s,a) when using a feedforward model. The statement is incorrect unless we use a linear function approximator as a model. However, this mistake does not affect much the quality of the paper.
- Using autoregressive models does not address aspects that are specific to the offline setting. Providing results for the online setting will be helpful for understanding whether autoregressive models should be favored in general for model-based reinforcement learning.


Recommendation

The reviewer votes for accepting the paper. The paper is well-written, the proposed extension is simple to implement and convincingly outperforms baselines on a variety of environments.


Notes
1. Appendix A.2 is identical to Section 5.1 of another submission [2].
2. The abbreviation FQE is used throughout the paper but expanded only in the appendix.


References

[1] Gulcehre, Caglar, Ziyu Wang, Alexander Novikov, Tom Le Paine, Sergio Gómez Colmenarejo, Konrad Zolna, Rishabh Agarwal et al. "Rl unplugged: Benchmarks for offline reinforcement learning." arXiv preprint arXiv:2006.13888 (2020).

[2] Anonymous. Benchmarks for deep off-policy evaluation. In Submission to International Conference on Learning Representations, 2021. URL https://openreview.net/forum?id=kWSeGEeHvF8. Under review.

---

> ### Author Response · Authors · 2020-11-20
> **Response to reviewer 1**
>
> Thank you for your valuable feedback! See our responses below:
>
> >The authors claim that “standard feedforward dynamics models assume that different dimensions of the next state and reward are conditionally independent given the current state and action”. In other words, p(s’,r|s,a) is claimed to be equal to p(s’_1|s,a) … p(s’_n|s,a) p(r|s,a) when using a feedforward model. The statement is incorrect unless we use a linear function approximator as a model. However, this mistake does not affect much the quality of the paper.
>
> Thanks for raising this point. In that sentence, we are referring to feedforward models that parametrize the next state as a multivariate Gaussian with a diagonal covariance structure (Chua et al. (2018); Janner et al. (2019)). In such a parameterization, each dimension of the next state is conditionally independent given the current state and action, independent of the class of functions used. We tried to make this clear in an updated abstract and introduction. Do our changes address your concern?
>
> >Using autoregressive models does not address aspects that are specific to the offline setting. Providing results for the online setting will be helpful for understanding whether autoregressive models should be favored in general for model-based reinforcement learning.
>
> We agree that extending our policy optimization results to the online RL setting is interesting. We opted to focus on the offline setting with a fixed dataset because it enables a more direct comparison between different model-based approaches. It allows us to study the interaction between negative log likelihood and expected return for dynamics models trained on the same dataset of interactions. Given the space constraints, we do not think it is feasible to include online RL experiments into the current paper, but we agree that this is an important direction for future work.
>
> We expanded on our definition of FQE in the main paper, thank you.

---

> > ### Comment · AnonReviewer1 · 2020-11-23
> > **Response to authors' feedback**
> >
> > Thank you for addressing the concerns in the review. Since the outlined weaknesses did not affect the final score much, the score remains unchanged and the reviewer recommends to accept the paper.

---

### Decision · Program_Chairs · 2021-01-07
**Final Decision**

**Decision:**

Accept (Poster)

**Comment:**

The paper is about the use of autoregressive dynamics models in the context of offline model-based reinforcement learning.
After reading the authors' responses and the other reviews, the reviewers agree that this paper has several strengths (well written, easy to follow, the approach is novel and simple to implement, the empirical evaluation is well executed and the results are reproducible) and it deserves acceptance.
The authors need to update their manuscript by keeping into considerations all the suggestions provided by the reviewers (clarifications and additional empirical comparisons).